# Interstellar Benzene Formation Mechanisms via Acetylene Cyclotrimerization Catalyzed by Fe^+^ Attached to Water Ice Clusters: Quantum Chemistry Calculation Study

**DOI:** 10.3390/molecules27227767

**Published:** 2022-11-11

**Authors:** Tatsuhiro Murakami, Toshiyuki Takayanagi

**Affiliations:** 1Department of Chemistry, Saitama University, Shimo-Okubo 255, Sakura-ku, Saitama 338-8570, Japan; 2Department of Materials & Life Sciences, Faculty of Science & Technology, Sophia University, 7-1 Kioicho, Chiyoda-ku, Tokyo 102-8554, Japan

**Keywords:** interstellar medium, astrochemistry, reaction dynamics, polycyclic aromatic hydrocarbons, transition-metal catalysis, quantum chemistry calculation

## Abstract

Benzene is the simplest building block of polycyclic aromatic hydrocarbons and has previously been found in the interstellar medium. Several barrierless reaction mechanisms for interstellar benzene formation that may operate under low-temperature and low-pressure conditions in the gas phase have been proposed. In this work, we studied different mechanisms for interstellar benzene formation based on acetylene cyclotrimerization catalyzed by Fe^+^ bound to solid water clusters through quantum chemistry calculations. We found that benzene is formed via a single-step process with one transition state from the three acetylene molecules on the Fe^+^(H_2_O)*_n_* (*n* = 1, 8, 10, 12 and 18) cluster surface. Moreover, the obtained mechanisms differed from those of single-atom catalysis, in which benzene is sequentially formed via multiple steps.

## 1. Introduction

Since the first spectroscopic detection of interstellar benzene (C_6_H_6_) [1], which is the simplest building block of polycyclic aromatic hydrocarbons in the interstellar medium [2,3,4], numerous theoretical and experimental studies have been performed to understand the benzene formation mechanisms under low-pressure and low-temperature interstellar conditions [5,6,7,8,9,10,11]. Many of these studies have proposed that benzene can be formed via barrierless processes, including ion–molecule, neutral radical–radical, and radical–molecule reactions in the gas phase. For example, Kaiser et al. [8] demonstrated that benzene molecules could be formed under single-collision conditions through the gas-phase barrierless reaction of an ethynyl radical (C_2_H) with trans-1,3-butadiene (C_4_H_6_) using a combined study of crossed molecular beam experiments and high-level quantum chemistry calculations, including statistical rate coefficient analyses. Recently, Habershon et al. [10] developed an automated reaction mechanism search algorithm from a given set of reactant and product inputs and applied it to the exploration of efficient interstellar benzene formation schemes. They confirmed that the reaction of C_2_H with trans-1,3-butadiene, initially proposed by the Kaiser group [8], is the most likely barrierless mechanism without human intuition and thus can be chosen as a good candidate for the interstellar benzene case. In addition, they identified that the reaction of C_2_H with two acetylene (C_2_H_2_) molecules is another favorable reaction for the formation of a C_6_H_5_ radical, which can be a precursor molecule of benzene [10]. It is worth mentioning that the gas-phase benzene formation process generally consists of multiple reaction pathways via several intermediates on the multidimensional potential energy surface.

In this work, we discuss the different formation mechanisms of interstellar benzene, which can be directly produced from three acetylene molecules catalyzed by atomic iron cations attached to water ice clusters, using quantum chemistry calculations. The direct synthesis of benzene from three acetylene molecules is generally called [2+2+2] acetylene cyclotrimerization [12,13,14,15]. Catalytic cyclotrimerization synthesis using various transition-metal-containing compounds and transition-metal clusters has been extensively studied in the field of organic chemistry [12,13,14,15,16] because non-catalytic cyclotrimerization is extremely slow even at high temperatures, owing to the large energy barrier associated with π-bond breaking [17,18,19]. Previous experimental [20,21,22,23,24] and theoretical [22,23,24,25,26,27,28] studies have shown that a single transition-metal atomic cation (or single neutral transition-metal atom) is sufficient to catalyze the acetylene cyclotrimerization reaction. For example, Shuman et al. [23] performed ion–molecule reaction experiments using the selected-ion flow tube technique and found that atomic Fe^+^ cations can efficiently catalyze the gas-phase acetylene cyclotrimerization reaction. Quantum chemistry calculations were also performed to understand the reaction mechanism and show that the overall reaction pathway consists of fewer steps than the gas-phase benzene formation mechanism [23,28]. Motivated by these studies, we theoretically investigated the acetylene cyclotrimerization reaction mechanism catalyzed by Fe^+^(H_2_O)*_n_* clusters, which may provide a possible model for the interstellar benzene formation on the water ice surface augmented by Fe^+^ cations.

Although iron is the sixth most abundant element on the basis of hydrogen, which is the most abundant element in astrophysical environments [29] and is a key element in life science [30], the role of iron and its compounds in the formation of interstellar molecules is not entirely understood in the astrochemical field. So far, only two molecules containing iron, FeO [31,32] and FeCN [33], have been spectroscopically detected in the interstellar media. Therefore, iron is a highly depleted element in interstellar and circumstellar environments. The current status of astrochemical observations has led to the speculation that iron must exist in other forms, such as iron nanoparticles and iron-containing dust particles [34], making the additional search for iron-containing molecules in the interstellar medium extremely necessary. We believe that quantum chemistry calculations would be useful for understanding the catalytic role of iron as the most abundant transition metal in the formation of various interstellar molecules. Recent quantum chemistry calculations related to the present work are also available in the literature [35,36,37,38,39].

## 2. Computational Details

All quantum chemistry calculations presented in this work were performed using the Gaussian09 software package [40] at the unrestricted B3LYP density functional theory (DFT) level. Our previous study showed that the B3LYP functional provides a reasonable spin-state order for the Fe-containing chemical systems [28,41,42,43]. Since the self-consistent field (SCF) calculation frequently converges to the solution with internal instability, we had to employ “stable = opt” calculations implemented in the Gaussian09 code. Using this option, the solution is forced to converge to the most stable SCF solution without internal instability. It is worth mentioning that it is very difficult to obtain smooth potential energy surfaces without this option, especially for transition-metal systems with open shells. It is important to include long-range dispersion interactions to discuss the reaction mechanisms quantitatively [44]. The GD3BJ option implemented in the Gaussian09 program was used to account for dispersion interactions [45]. Most of the calculations were performed using the def2-SVPP basis set. However, the def2-TZVP basis set was also used for small cluster systems to understand the basis set effect.

## 3. Results and Discussion

Figure 1 shows the cyclotrimerization pathway (solid black line) from the Fe^+^(C_2_H_2_)_3_ reactant complex to the Fe^+^C_6_H_6_ product complex calculated at the B3LYP(D3BJ)/def2-SVPP level as a function of the reaction path length, which corresponds to the distance calculated using mass-weighted coordinates. Note that this process occurs entirely on the potential energy surface with a quartet spin state [28]. In this case, the overall cyclotrimerization pathway consists of two intrinsic reaction coordinate (IRC) calculation results. These two IRC potential curves are merged into a single curve in this plot, where the horizontal coordinate origin is set to the transition state structure (denoted as TS_1_) of the first IRC path, and the energy is measured from the energy level of the optimized Fe^+^(C_2_H_2_)_3_ reactant complex. In this reactant complex, it was found that three acetylene molecules were equivalently bound to the Fe^+^ cation through a charge-π interaction (and a small contribution of the *d*-π bonding interaction) with *D*_3_ symmetry. Thus, the π-bond in each acetylene molecule is weakened, indicated by the non-linear structures of the three acetylene molecules shown at the left-top in Figure 1. The first step from the reactant complex to the intermediate (INT_1-2_) formation occurs through the transition state (TS_1_), which corresponds to the dimerization reaction (the first CC σ-bond formation) between the two acetylene molecules. The activation energy for TS_1_ originates from the breaking of the acetylene π-bond. Interestingly, one acetylene molecule mostly acts as a spectator in this first reaction pathway, and the orientation of this nonreactive acetylene was found to be nearly perpendicular to the C-C-C-C plane of the dimerization product, INT_1-2_. The transition state structure (TS_2_) of the second IRC path corresponding to the benzene production process shows the expected structural feature, where all six carbon atoms are located approximately on a single plane with two newly formed CC σ-bonds at distances of 2.7–2.8 Å. Additionally, the energy barrier of the TS_2_ structure measured from the INT_1-2_ intermediate should be partially associated with the rotational barrier of the third acetylene molecule for the acetylene molecule to adopt a nearly planar structure. Thus, the corresponding energy barrier measured from INT_1-2_ is large (>20 kcal/mol). Note that the reaction process has a large exothermicity in order to form stable benzene.

Figure 1 also shows the results for the acetylene cyclotrimerization process catalyzed by the complex of Fe^+^ and a solid water cluster consisting of eight water molecules. Here, we chose to employ the most stable water cluster structure with *D*_2*d*_ symmetry [46] as an initial optimization structure to avoid structural changes during the acetylene cyclotrimerization reaction. The reaction pathway was calculated for the quartet spin state because this spin state was the most stable (see below). In this case, the Fe^+^ cation was preferentially bound to the O atom of the water molecule, which did not have a dangling OH bond. The (C_2_H_2_)_3_-Fe^+^(H_2_O)_8_ reactant complex also has a charge-π interaction similar to the single-atom Fe^+^ case, including the contribution of the *d*-π bonding. The feature of the bonding interaction is described in the Kohn–Sham orbitals shown in Appendix A. Interestingly, three acetylene molecules are bound to Fe^+^, forming an approximately planar structure in the optimized (C_2_H_2_)_3_-Fe^+^(H_2_O)_8_ reactant complex. This is in contrast with the case of the single-atom Fe^+^ catalysis. More interestingly, we also found that the acetylene cyclotrimerization reaction from the reactant complex to the benzene complex occurred via a single IRC pathway, including only one transition state structure (TS_W8_). The intermediate structure observed in the single-atom Fe^+^ case is completely missing for the Fe^+^(H_2_O)_8_ catalysis case. In this case, the TS_W8_ has a nearly planar (C_2_H_2_)-(C_4_H_4_)-Fe^+^(H_2_O)_8_ structure while a remaining C_2_H_2_ is perpendicular to the C_4_H_4_ of TS_1_ for the single-atom Fe^+^ case. Interestingly, we found that the C-C σ-bonding character between the remaining C_2_H_2_ and C_4_H_4_ is seen already at TS_W8_. This should be the main reason for the missing barrier. The C-C σ-bonding orbital character is shown in Appendix A. In addition, the energy barrier measured from the reactant energy level was reduced to 12.9 kcal/mol, which was much lower than that of the single-atom Fe^+^ (19.7 kcal/mol). This indicates that the Fe^+^ cation bound to water acts as a more efficient catalyst than the bare atomic Fe^+^ cation. Notice that the potential energy curve with the fixed (H_2_O)_8_ structure on the IRC pathway also describes the single-step cyclotrimerization (see Appendix A).

To further understand the structural changes in the acetylene cyclotrimerization reaction catalyzed by the Fe^+^(H_2_O)_8_ cluster, selected atomic distances were plotted and are shown in Figure 2 as a function of the same reaction path length defined in Figure 1. Figure 2a shows the four Fe-C*_i_* (*i* = 1, 2, 3, and 6) distances, while three CC bond distances are plotted in Figure 2b. The three CC distances gradually decrease up to *s* ≈ –4 from *s* ≈ –10, where *s* denotes the reaction path length. After this point, the C_4_-C_5_ distance decreases further, indicating the formation of a CC σ-bond. Once the C_4_-C_5_ σ-bond is formed, singly occupied p-orbitals are isolated at the C_3_ and C_6_ sites. These two p-orbitals subsequently attract the π-bond of the third acetylene molecule (see Appendix A). In fact, after the transition state (at *s* = 0), the C_1_-C_6_ and C_2_-C_3_ distances decrease gradually, leading to the formation of two CC σ-bonds. Over *s* = 10, three CC bond distances barely change with a gradual energy stabilization to the benzene complex depicted in Figure 1. During this cyclotrimerization process, a change in the Fe-C*_i_* distance can be seen from the results presented in Figure 2a, indicating a change in the coordination structure during the cyclotrimerization process. From *s* ≈ –10 to 10, four Fe-C coordination distances shrink slightly (1.9–2.0 Å) to assist to form benzene. It could be found that the benzene complex is more stable at the long Fe-C coordination bond, where the distances are 2.3–2.4 Å over *s* = 10, shown in Figure 1 and Figure 2.

As mentioned above, the quartet state is the lowest spin state in this reaction. In order to confirm it, we investigated the effect of other spin states, namely doublet and sextet, because these states are energetically close to the quartet spin state [23,24,28]. Figure 3 shows the potential energy profiles for the sextet and doublet spin states along the IRC structures calculated for the quartet spin state. No crossing points can be observed for these three potential energy curves, indicating that acetylene cyclotrimerization catalyzed by the Fe^+^(H_2_O)_8_ cluster occurs exclusively on the quartet potential energy surface.

Next, we studied the effect of the number of water molecules on the catalytic properties for the acetylene cyclotrimerization process. Figure 4 shows the potential energy profiles calculated at the B3LYP/def2-SVPP level for the reactions catalyzed by the Fe^+^(H_2_O), and Fe^+^(H_2_O)_10_, and Fe^+^(H_2_O)_8_ complexes. The stationary-point structures on the potential energy surfaces are shown in Figure 5, and their Cartesian coordinates and zero-point energies are presented in the Appendix A. For the Fe^+^(H_2_O)_10_ complex, we chose to employ the most stable pentagonal structure as the initial optimization structure for the (H_2_O)_10_ moiety [47]. In all cases, the relative energy levels of the (C_2_H_2_)_3_-Fe^+^(H_2_O)*_n_* (*n* = 1, 8, and 10) complexes are defined as zero in Figure 4; therefore, the results are consistent with those in Figure 1. It is interesting to note that single-step cyclotrimerization occurs even for *n* = 1, indicating that the Fe^+^-H_2_O complex acts as an efficient catalyst for acetylene cyclotrimerization. The barrier height measured for the (C_2_H_2_)_3_-Fe^+^(H_2_O)*_n_* (*n* = 1, 8, and 10) complexes decreased slightly with increasing *n*. For the Fe^+^-H_2_O complex, we also calculated the IRC path using the larger def2-TZVP basis set. We found that the obtained result was similar to that of the def2-SVPP case, as shown in Appendix A. We have also performed similar calculations for the Fe^+^(H_2_O)_12_ (with a hexagon structure) and Fe^+^(H_2_O)_18_ (consisting of multiple cubic structures). The results are presented in Appendix A. These results indicate that the barrier height is not largely dependent on the water cluster size for *n* ≥ 8 nor the detailed cluster structure.

To understand the astrochemical implications of the present study, we show the energy levels of the sum of the (C_2_H_2_)_2_-Fe^+^(H_2_O)*_n_* complex and free C_2_H_2_ energies. It should be noted that in the optimized (C_2_H_2_)_2_-Fe^+^(H_2_O)*_n_* complex, two acetylene molecules are bound to Fe^+^ through charge-π and *d*-π interactions. The results in Figure 4 indicate that the energy levels of the asymptotic C_2_H_2_ + (C_2_H_2_)_2_-Fe^+^(H_2_O)*_n_* complex are higher than those of the TS_w*n*_ because of the strong, attractive interaction between C_2_H_2_ and (C_2_H_2_)_2_-Fe^+^(H_2_O)*_n_*. Therefore, the acetylene cyclotrimerization depicted in Figure 4 is a barrierless exothermic reaction, at least for *n* = 1, 8, and 10, within the most stable water cluster structure framework. Assuming that the attractive potential energy between C_2_H_2_ + (C_2_H_2_)_2_-Fe^+^(H_2_O)*_n_* is employed to surmount the TS_w*n*_ barrier, benzene can be efficiently formed. However, if this entrance potential energy is fully dissipated into other vibrational modes, especially in the water cluster during the formation of the optimized (C_2_H_2_)_3_-Fe^+^(H_2_O)*_n_* complex, the benzene formation process would be very slow because the barrier height measured from the complex is still greater than 10 kcal/mol, which is much higher than the thermal energy.

Finally, we discuss the dynamical aspects of the acetylene cyclotrimerization reaction catalyzed by the Fe^+^(H_2_O)*_n_* complex. The dynamical aspect should be carefully examined because non-IRC dynamics frequently occur in complicated reaction systems [48,49]. In this case, the reaction product cannot be identified only from the static IRC calculations. It is possible that a large reaction exothermicity can provide additional kinetic energy to the atoms in the reaction system, leading to a deviation from the IRC pathway. In addition, it is important to understand the energy transfer dynamics because the acetylene cyclotrimerization process has a large exothermicity associated with stable benzene formation. Motivated by this, we performed molecular dynamics calculations starting from the transition state structure with a Born–Oppenheimer molecular dynamics (BOMD) keyword implemented in the Gaussian09 program [40], where energy gradients were employed to solve the equations of motion of the nucleus. The BOMD calculations were only performed for the Fe^+^(H_2_O)_8_ cluster. We added a total kinetic energy of 1 kcal/mol for the atoms in the Fe^+^(C_2_H_2_)_3_ moiety. If the trajectory moved toward the (C_2_H_2_)_3_-Fe^+^(H_2_O)_8_ reactant complex region, the calculation was terminated, whereas if the trajectory moved toward the benzene formation direction, the calculation was performed up to *t* = 1 ps. The time increment was set at Δ*t* = 0.5 fs. Since BOMD calculations generally require a large computational time, even at the B3LYP/def2-SVPP level, we only calculated six trajectories showing benzene formation in this study. Here, it is worth noting that the dissociation of benzene from the Fe^+^(H_2_O)_8_ cluster was not observed in all the six trajectories (see below). A typical example is presented in Figure 6, where the potential energy is plotted as a function of simulation time in Figure 6a, and the kinetic energies for the C_6_H_6_, Fe^+^, and (H_2_O)_8_ moieties are plotted in Figure 6b. In this trajectory, a benzene molecule is formed around t ≈ 0.2 ps. Subsequently, a large amount of exothermic energy was distributed in the vibrational energy of benzene. We can also observe the energy transfer from the benzene vibration to the water cluster. At *t* = 1 ps, approximately 20 kcal/mol of energy was transferred to the atomic kinetic energy (vibrational energy) of the (H_2_O)_8_ cluster, leading to the breakage of a few hydrogen bonds. As mentioned previously, we did not observe the dissociation of benzene from the cluster within 1 ps of simulation time, presumably due to the large binding energy of ~50 kcal/mol between C_6_H_6_ and Fe^+^(H_2_O)_8_ (see Figure 4). The results for the other five trajectories were found to be very similar, as shown in Appendix A.

## 4. Conclusions and Future Directions

In this study, we performed DFT-level quantum chemistry calculations for the acetylene cyclotrimerization reaction catalyzed by the Fe^+^(H_2_O)*_n_* cluster to understand the possible benzene formation mechanisms in the interstellar medium. Interestingly, we found that benzene was concertedly formed via a single reaction step, including only one transition state structure from three acetylene molecules. We confirmed that the obtained reaction path was robust to benzene formation by calculating the classical trajectories starting from the transition state structure. This single-step scheme is in high contrast to the bare Fe^+^ catalysis case, in which benzene is sequentially formed via two steps, consisting of a single CC σ-bond formation and the two subsequent CC σ-bond formations. It should be emphasized that the benzene formation barrier height for the Fe^+^(H_2_O)*_n_* cluster catalysis was reduced compared to that for the single-atom Fe^+^ case. Nevertheless, once three acetylene molecules were attached to the Fe^+^(H_2_O)*_n_* cluster and their structure was fully relaxed, the reaction pathway reached a substantial barrier to form benzene. The barrier height was found to be significantly larger than the thermal energy. We plan to extend the present quantum chemistry calculation study to C_2_ and C_2_H radicals, which are known to exist in the interstellar medium. In this case, we expect that the benzene precursor molecules, such as C_6_H*_n_* (*n* = 0–5), can be formed via cyclization reaction mechanisms with much lower barriers than the acetylene cyclotrimerization reactions.

## Figures and Tables

**Figure 1 molecules-27-07767-f001:**
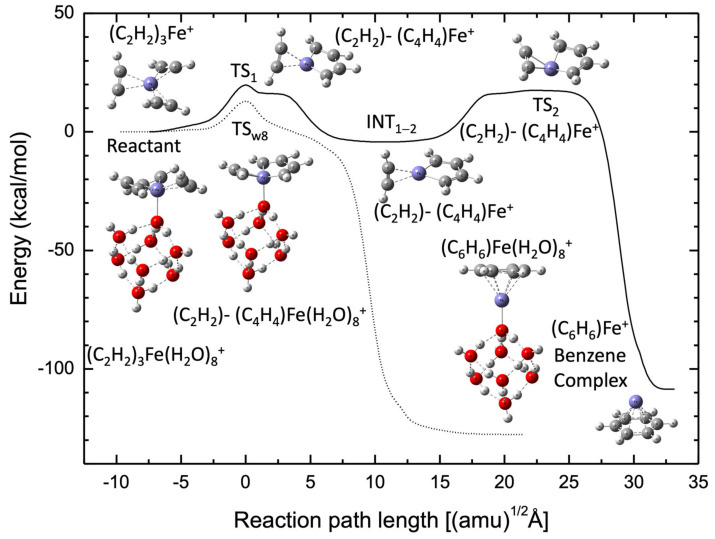
Potential energy profiles for the acetylene cyclotrimerization reactions catalyzed by Fe^+^ and Fe^+^(H_2_O)_8_ plotted as a function of the reaction path length obtained at the B3LYP(D3BJ)/def2-SVPP DFT level.

**Figure 2 molecules-27-07767-f002:**
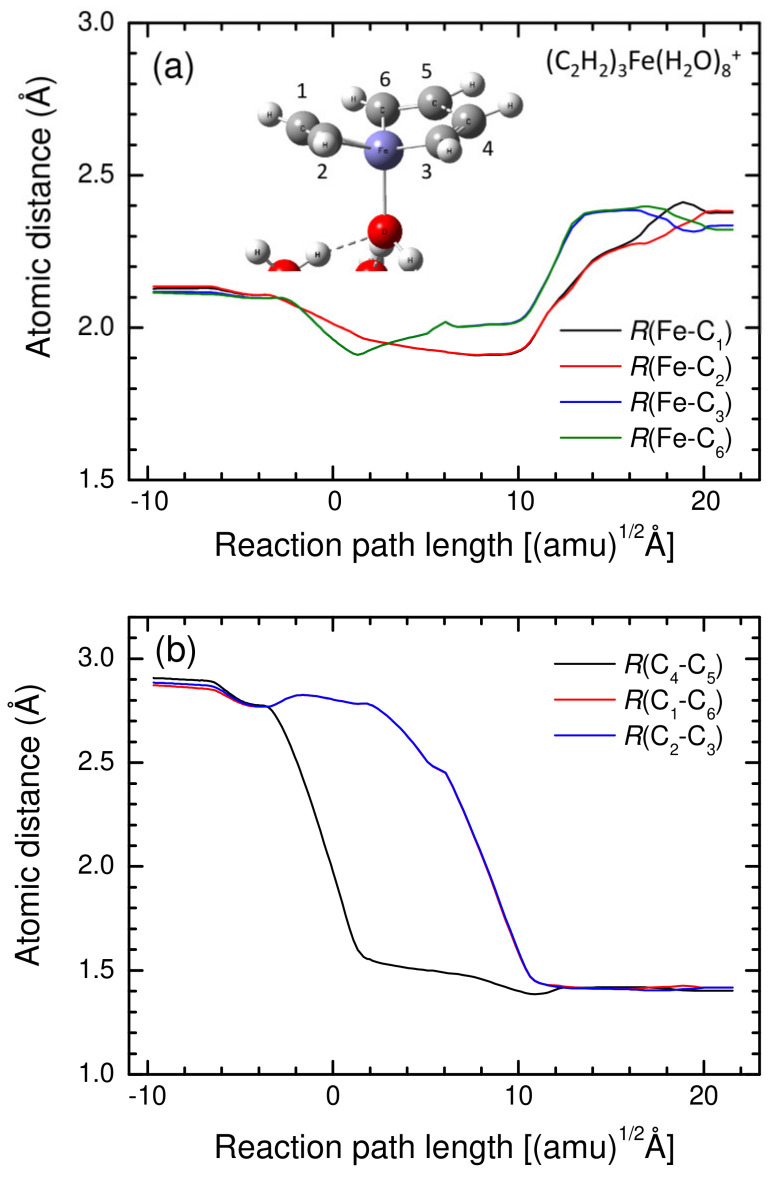
Key atomic distances for (**a**) Fe-C and (**b**) C-C along the acetylene cyclotrimerization reaction catalyzed by the Fe^+^(H_2_O)_8_ cluster plotted as a function of the reaction path length defined in Figure 1.

**Figure 3 molecules-27-07767-f003:**
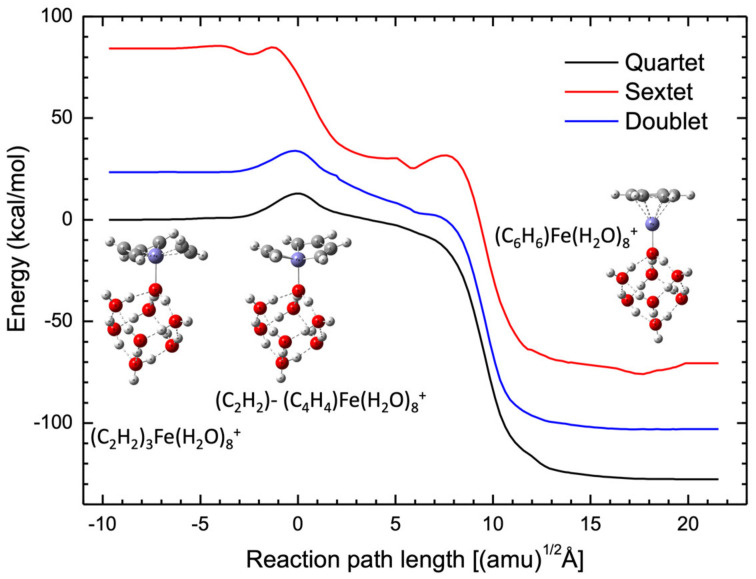
Comparison of potential energy profiles with different spin states for the acetylene cyclotrimerization reaction catalyzed by the Fe^+^(H_2_O)*_n_* cluster. The blue, black, and red lines represent the doublet, quartet, and sextet states, respectively.

**Figure 4 molecules-27-07767-f004:**
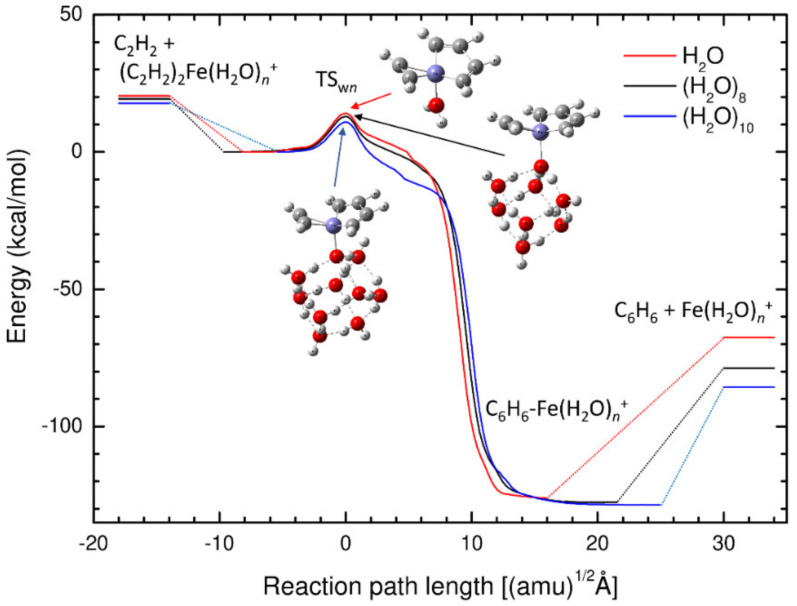
Comparison of the potential energy profiles for the acetylene cyclotrimerization reactions catalyzed by three different Fe^+^(H_2_O)*_n_* (*n* = 1, 8, and 10) clusters.

**Figure 5 molecules-27-07767-f005:**
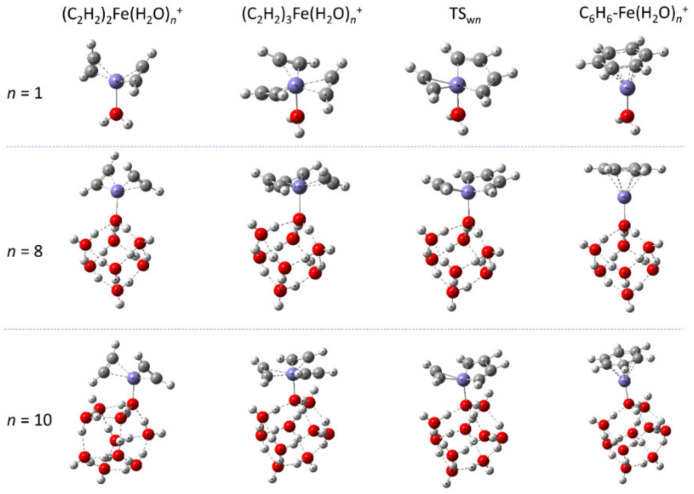
Stationary-point structures on the potential energy surfaces for the acetylene cyclotrimerization reactions catalyzed by three different Fe^+^(H_2_O)*_n_* (*n* = 1, 8, and 10) clusters.

**Figure 6 molecules-27-07767-f006:**
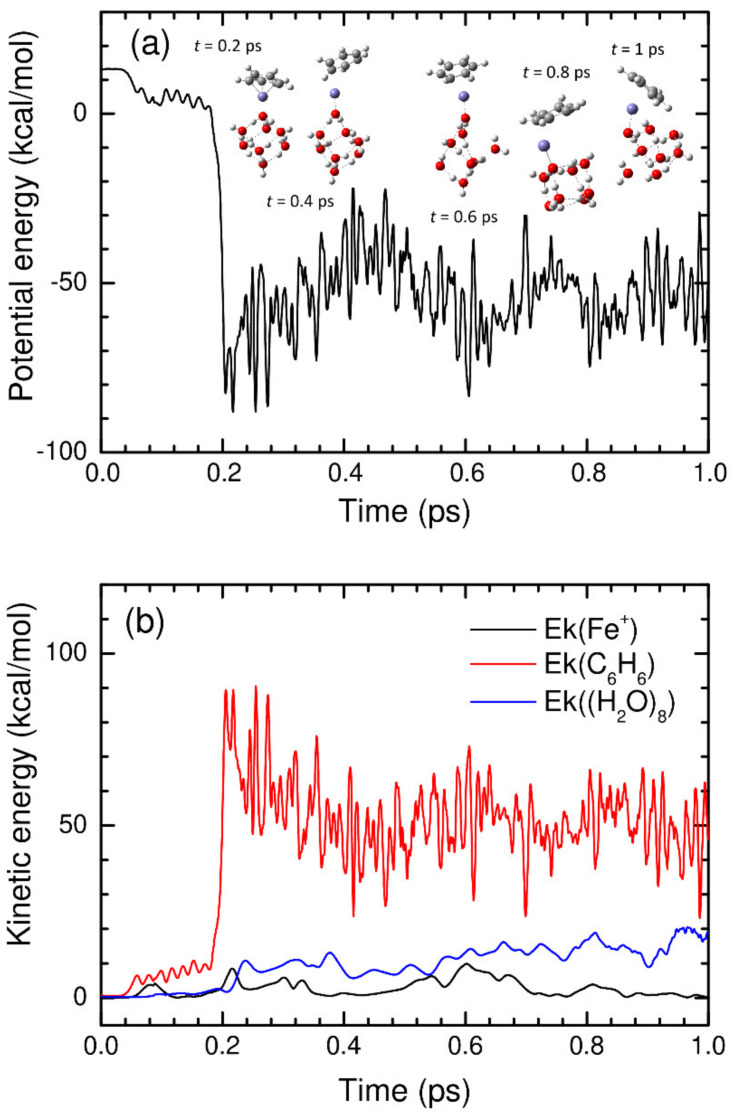
(**a**) Potential energy and (**b**) atomic kinetic energies obtained from the BOMD trajectory starting from the transition state structure with an excess energy of 1 kcal/mol plotted as a function of simulation time.

## Data Availability

Not applicable.

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
