# Peer review of "Interstellar Benzene Formation Mechanisms via Acetylene Cyclotrimerization Catalyzed by Fe+ Attached to Water Ice Clusters: Quantum Chemistry Calculation Study"

_molecules, 2022, doi:10.3390/molecules27227767_

Round 1

Reviewer 1 Report

The authors report an interesting computational work on catalysis by Fe+ ion on a corner of a hypothetical (H2O)8 cube as a model for ice.  The results are convincing and worth publication.  The reviewer has a few comments as the following:

(1) The definition of catalysis needs to be clarified on the reactions that are studied here.  The reactions (shown in Figure 1) are actually formation of benzene from three acetylene molecules that are BOUND throughout the reaction.  To claim that Fe+ ion is catalyzing the reactions, one must start with free acetylene molecules and produce free benzene.  Since the molecules are always bound to Fe+, the reactions are not catalyzed reactions but internal reactions.  For example, the chemical equation in line 201, “C2H2 + (C2H2)2-Fe+(H2O)n → C6H6 (benzene) + Fe+(H2O)n” is not correct, as it indicates that C2H2 and C6H6 are free molecules, although they are not.

(2) The authors describe that the acetylene molecules are bound to Fe+ ion through charge-pi interaction.  But, no evidence is given.  It is customary to think that transition metal ions such as iron have strong covalent interactions with pi systems.  The most famous example would be ferrocene, Fe(C5H5)2.  It would be helpful to examine the contribution of the pi-orbitals of the organic molecules to the Kohn-Sham orbitals of Fe+ orbitals near HOMO.  

(3) From computation, it was found that there is no transition state from Fe+(C4H4)(C2H2)(H2O)8 to Fe+(C6H6)(H2O)8.  This is rather unusual.  Additional comments on this would be helpful.

(4) The role of the (H2O)8 is not clear.  Does it provide an electronic effect for the changed reaction path or simply act as an steric object that forces the organic molecules to be arranged in near planar geometry.  With the reaction path of (C2H2)3Fe+(H2O)8, calculation of the potential energy curve with “fixed” atomic positions after taking out (H2O)8 is necessary to discern the effects.  

Author Response

See an attached file.

Reviewer 2 Report

In this manuscript, the authors studied different mechanisms for interstellar benzene formation based on acetylene cyclotrimerization catalyzed by Fe+ bound to solid water clusters through DFT method.  The results obtained are very helpful for the synthesis of benzene precursor molecules. Moreover, the data in this manuscript is very rich, the discussion is perfect and the expression is clear. Therefore, this manuscript can be accepted in its current form.

Author Response

See an attached file.

Reviewer 3 Report

A large number of theoretical and experimental studies have been carried out to understand the formation mechanism of benzene in the interstellar medium. In this paper, the authors reported another possible mechanism for interstellar benzene formation based on acetylene cyclotrimerization catalyzed by Fe+ bound to solid water clusters through quantum chemistry calculations. The text is generally well written, and the figures and tables are helpful to a reader’s understanding. Most importantly, the authors apply solid reasoning to explain the observed trends in the data. This paper should add to our understanding of the formation mechanism of benzene in the interstellar medium, so publication in Molecules can be recommended after attention to the pervading errors and infelicities of English language and style.

Author Response

See an attached file.

Reviewer 4 Report

This article thoroughly reported a theoretical study for interstellar benzene formation mechanisms via acetylene cyclotrimerization by Fe+ attached to water ice clusters. By using quantum chemistry calculations, the authors illustrated the acetylene cyclotrimerization to benzene pathway in details from multiple aspects, i.e., the potential energy diagram, structural changes, effects of iron spin states, number of water molecules, and the dynamical features.

Overall, this research is well organized and designed and the results are helpful to the astrochemistry field, especially for the understanding of acetylene cyclotrimerization and catalysis by clusters. It is recommended to be accepted and published on molecules.

It will be better for the readability if the corresponding reactants, intermediates, products in the potential energy diagrams (Figure 1, Figure 2, Figure 3) can be labeled with chemical formula.

Author Response

See an attached file.

Round 2

Reviewer 1 Report

The reviewer suggests the publication of the revision.